# Delta T, a Useful Indicator for Pharmacy Dispensing Data to Monitor Medication Adherence

**DOI:** 10.3390/pharmaceutics14010103

**Published:** 2022-01-02

**Authors:** Pascal C. Baumgartner, Bernard Vrijens, Samuel Allemann, Kurt E. Hersberger, Isabelle Arnet

**Affiliations:** 1Pharmaceutical Care Research Group, Department of Pharmaceutical Science, University of Basel, 4051 Basel, Switzerland; s.allemann@unibas.ch (S.A.); kurt.herberger@unibas.ch (K.E.H.); isabelle.arnet@unibas.ch (I.A.); 2AARDEX Group, Avenue de la Gare 29, 1950 Sion, Switzerland; bernard.vrijens@aardexgroup.com

**Keywords:** medication adherence, compliance, pharmacy claims, measures, cluster analysis

## Abstract

Introduction: Calculating patients’ medication availability from dispensing or refill data is a common method to estimate adherence. The most often used measures, such as the medication possession ratio (MPR), average medication supplies over an arbitrary period. Averaging masks the variability of refill behavior over time. Goal: To derive a new absolute adherence estimate from dispensing data. Method: Dispensing histories of patients with 19 refills of direct oral anticoagulants (DOAC) between 1 January 2008 and 31 December 2017 were extracted from 39 community pharmacies in Switzerland. The difference between the calculated and effective refill day (ΔT) was determined for each refill event. We graphed ΔT and its dichotomized version (dΔT) against the MPR, calculated mean ΔT and mean dΔT per refill, and applied cluster analysis. Results: We characterized 2204 refill events from 116 DOAC patients. MPR was high (0.975 ± 0.129) and showed a positive correlation with mean ΔT. Refills occurred on average 17.8 ± 27.9 days “too early”, with a mean of 75.8 ± 20.2 refills being “on time”. Four refill behavior patterns were identified including constant gaps within or at the end of the observation period, which were critical. Conclusion: We introduce a new absolute adherence estimate ΔT that characterizes every refill event and shows that the refill behavior of DOAC patients is dynamic.

## 1. Introduction

Electronic healthcare data (EHD) represent a non-intrusive, low-cost data source for the retrospective estimation of medication adherence in large populations [1,2] and can be used prospectively for adherence management [3,4]. Prescribing, dispensing, or claims data allow calculation of a patient’s medication availability over a defined observation period. Different calculation methods exist, which mostly differ regarding one of three characteristics: the distribution of the medication adherence variable (continuous or dichotomous), the number of refill intervals (single or multiple), and the consideration of gaps [5]. The most used measures are continuous multiple interval measures of medication availability (CMA), such as the medication possession ratio (MPR) or the proportion of days covered (PDC). They represent the proportion of days’ supply during the observation period [6]. For any calculation method, a record of each medication event and the duration of the supply (elaborated from the refill data) are mandatory [7]. Based on these variables, the numerator can be operationalized either as the sum of all day’s supplies (MPR) or as the days covered with supply (PDC), and the denominator is the length of the observation period. The calculated rates are usually interpreted by setting a threshold to distinguish adherent from nonadherent patients [8,9,10]. The most often used threshold is 0.8 [11], while 0.95 is often applied for medicines requiring strict adherence such as direct oral anticoagulants (DOAC) [12]. The CMA is a single number with several limitations. The main limitation is the strong dependence on the defined observation window that delineates the included medication events. As a consequence, different medication adherence rates may be obtained with the same patient data and may misclassify a patient as a non-adherer. Second, CMAs are aggregate measures, and the prediction of patients’ refill behavior from CMAs is undifferentiated. As an example, CMAs cannot differentiate between a patient with a low implementation rate, and a patient with a high implementation rate who has discontinued their therapy, mainly because most EHD do not allow identifying precisely the time point of discontinuation [13]. Further, an aggregate estimate obscures the variability of refills over time i.e., the dynamic of medication adherence. Such inaccuracy deviates from consensus-based guidelines’ advice on best practice in defining medication adherence (ABC-Taxonomy) [14,15] or reporting of empirical studies (EMERGE-guidelines) [16]. According to these new recommendations, medication adherence research should specify the medication adherence phases under scrutiny, that is, initiation, implementation, or discontinuation. Some new approaches have been recently developed on how to display the temporal refill patterns, in other words how to characterize a continuous single interval measure of medication availability (CSA). One variation involves calculating the PDC over two refill intervals (so-called time-varying PDC [17]). Another approach is to use shorter and potentially overlapping observation windows (so-called sliding windows) to obtain more precise statements about the patient’s refill behavior [18]. To our knowledge, no study has analyzed the absolute relationship between single refill events from a patient population to characterize patient behavior, and the potential of this method has not yet been assessed. We hypothesize that opposite to MPR, the characterization of every single refill event allows depicting a patient’s refill behavior over time. The usefulness of this new approach consists in delineating the dynamic of medication adherence. Our goal was to derive a new absolute adherence estimate from dispensing data.

## 2. Methods

### 2.1. Development of the New Estimate Delta T (ΔT)

We used the nomenclature for CSA and CMA proposed by Steiner et al. [5], the ABC-Taxonomy [15], the TEOS Framework [19], and the standardized elements according to Arnet [20] to develop the new estimate (see Appendix A). We specified that the new estimate describes every refill event (R_n_) during the implementation phase of pharmacotherapy and defined implementation as consecutive dispenses with no gap of more than 182 days (=6 months). We assumed that every dispensing record includes exact single event dates and further variables so that the number of days’ supply (A_n_) and the refill interval (B_n_) can be calculated [7]. Two dispensing events at least are needed. The new estimate Delta T (ΔT) is calculated at each refill event (R_n_) as the difference between the number of days of medicine previously supplied (A_n_) and the number of days in the corresponding refill interval (B_n_). Positive values and zero (ΔT ≥ 0) indicate the number of days the patient has a sufficient supply at the refill event Rn, and negative values (ΔT < 0) indicate days without supply or “gaps” before the next refill event (see Figure 1). We defined a dichotomized form of ΔT (dΔT) with 1 for “on time” refill events (ΔT ≥ 0; the patient obtained a refill before running out of supply) and 0 for “too late” refill events (ΔT < 0; the patient had not enough supply to cover the period until the next refill). In the case of oversupply, the number of days’ supply is carried over to the next number of days’ supply (A_n_ + 1), thus assuming patients will terminate the oversupply before using the new supply. This approach should prevent the underestimation of medication adherence at the patient level over time [21,22].

### 2.2. Data Source

We selected real-life dispensing data from TopPharm pharmacies in Switzerland. From the 130 independent pharmacies of the group, 39 (30%) agreed to participate in the study. We selected the direct oral anticoagulants (DOAC; ATC Codes: B01AF01, B01AF03, B01AF02, B01AE07) for their non-forgiving property that requires strict medication adherence. We extracted dispensing histories of patients with at least two dispenses (that is, one fill and one refill) between 1 January 2008 and 31 December 2017 (10 years coverage). For every case, we obtained patient characteristics: year of birth, gender, zip code, number of further medicines (=unique ATC Codes in the first 12 months after the first dispensing event of any DOAC in the follow-up period), and dispensing characteristics (date of the medication event, ATC code, the strength of the medicine supplied, quantity dispensed, prescribed daily dosage). We assumed that the higher the number of refills, the more likely it is that medication possession and consequently MPR will be high. Simultaneously, we expect variations in refill behavior that are sufficiently marked to be detected. Therefore, we selected all patients from pharmacy databases with at least 20 consecutive dispenses of DOAC. This should guarantee a theoretical refill period greater than 1.5 years, extrapolated from a package of 30 tablets that lasts for one month. Approval for the data export and extraction was obtained from the Ethics Committee of Northwestern Switzerland (EKNZ Nr. 2018-01490, 11 September 2018).

### 2.3. Analytical Procedure and Statistical Analysis

We calculated ΔT and dichotomized ΔT (dΔT) for each refill. The MPR was calculated for each patient according to Vollmer et al. [23] and the values were dichotomized following the common 80% threshold [11]. As recommended to fully understand the structure of the estimates [9], we characterized and graphed the distribution of the MPR, average ΔT, and average dΔT for each patient; and characterized the population by the descriptive statistics of the mean value, standard deviation, the median, interquartile range (IQR), maximum (max ΔT) and minimum values (min ΔT). We graphed mean ΔT and mean dΔT against the corresponding MPR value of each patient to investigate how the average refill time corresponds to the medication availability, and computed Spearman’s rank correlation to assess the relationship between MPR and ΔT. We plotted ΔT over the 19 refills for three illustrative patients to visualize the dynamic of different refill behaviors. The three illustrative patients showed a “perfect” MPR of 1; an MPR of 0.95 [12] and an MPR below 0.8 [11]. For each patient, median ΔT (IQR), mean ΔT ± standard deviation, max ΔT, min ΔT, the corresponding range (=max ΔT–min ΔT), dΔT [%], and the sum of days without supply (=sum of negative ΔT) were calculated. To see an overall trend in the refill behavior with increasing refill number in the population, we calculated mean ΔT per refill over all patients, the percentage of patients per refill who were “on time” (ΔT ≥ 0), and computed Spearman’s rank correlation between ΔT and increasing refill number. To visualize the dynamics of different refill behaviors in the population, we plotted heat maps of ΔT and dΔT. We applied cluster analysis to classify patients into different refill behavior patterns. For this purpose, we used hierarchical cluster analysis with the dichotomized Euclidian method and the linkage method furthest distance neighbor measure. The data were analyzed with the Statistical Package for the Social Sciences (SPSS; Version 25.0 IBM Corporation, Armonk, NY, USA), or Microsoft Excel (Microsoft Office Home and Student 2016, Microsoft Corporation, Redmond WA, USA), or Tableau Desktop Professional Edition Version (2019.3.0, Tableau Software, Seattle, WA, USA). Heat maps were generated with Tableau Desktop Professional Edition Version (2019.3.0, Tableau Software, Seattle, WA, USA).

## 3. Results

### 3.1. Study Population

A total of 2919 pharmacy dispensing histories of patients were extracted of whom 116 (4%) patients had at least 20 consecutive dispenses (19 refills) of DOAC, corresponding to a total of 2204 refill events. At the first DOAC dispense, patients were on average 72.01 ± 10.91 years old with 54.2% women and were additionally obtaining a mean of 15 ± 7.84 different medicines during the next 12 months. The period for refilling 19 times the DOAC was on average 3.15 ± 1.28 years (range: 0.74 to 5.39 years). The mean supply duration was 59.7 ± 25.6 days (range: 14.7 to 98.0 days). The most often dispensed DOAC were rivaroxaban (69.3%) followed by dabigatran (15.7%) and apixaban (15%). A switch between DOAC was rare (16.1%) and was mostly from rivaroxaban to apixaban (54.5%).

### 3.2. Mean Delta T and Dichotomized Delta T

Overall, refills occurred on average 17.8 ± 27.9 days “too early” (see Figure 2A). The patients were on average 14 out of 19 times “on time” to refill their DOAC (mean dΔT: 75.8 ± 20.2%, see Figure 2B). A positive mean ΔT indicating DOAC oversupply was observed for 95 patients (81.1%).

### 3.3. Comparison with the Medication Possession Ratio

Mean MPR for DOAC was high with 0.975±0.129 and 104 (89.7%) patients with a MPR above 0.8 (see Figure 2C). There was a positive correlation between the two variables MPR and mean ΔT (r(114) = 0.778, *p* = 0.001, see Figure 2D).

### 3.4. Individual Refill Pattern of Three Illustrative Patients

Large fluctuations of ΔT over the 19 refills were observed for the three illustrative patients (see Figure 3) including refills that were “too late” (between 3 refills for the patient with an MPR of 1 and 7 refills for the patient with MPR of 0.78). The number of gaps in supply, that is, a negative ΔT, ranged from 12 (for the patient with MPR of 1) to 84 days (for the patient with an MPR of 0.78; see Table 1) with a different temporal pattern. The patient with an MPR of 0.78 presented gaps at the beginning and the end (refill number 6, 18), the patient with an MPR of 0.95 in the middle (refill number 10, 11), and the patient with an MPR of 1 at the end (refill number 15, 17) of the refill period.

### 3.5. Refill Trend in the Population

There was a positive correlation between the refill number and mean ΔT (r (114) = 0.950, *p* = 0.001, see Figure 4). The values increased from one refill to the next by approximately 20% with an increase by 27.2 days from the first to the last dispense (mean ΔT_1_: 4.9 days vs. mean ΔT_19_: 32.1 days). The average percentage of patients that were “on time” fluctuated from refill to refill between 69.8% (ΔT_11_) and 83.6% (ΔT_4_) with no observed trend over time.

### 3.6. Refill Groups within the Population

The 2204 individual DOAC refill events were visualized in heat maps that replicated all single ΔT (19 refills from 116 patients) with the color code green for refill events “on time” (ΔT ≥ 0) and red for refill events “too late” (ΔT < 0). The picture obtained was standardized with a color gradient between −150 days (red) and +150 days (green, see Figure 5A). When values were dichotomized (dΔT), then the dominance of the green color appeared (see Figure 5B). When the cluster analysis was applied to dΔT, four different patterns were differentiated (see Figure 5C and Table 2). We characterized the largest group (*n* = 71; 61.2%) as patients who consistently refilled “on time”, followed by erratic pattern (*n* = 29; 25.0%), gaps in the middle or at the end of the refill period (both with *n* = 8; 6.9%).

## 4. Discussion

In our data set comprising 116 patients with 19 refill events over up to 5.4 years, we were able to calculate ΔT for 2204 refill events for DOAC and showed trends of refill behavior that enabled us to define four different groups of refill patterns. In this highly selective sample with high MPR values and a small scatter, ΔT permitted a more differentiated characterizing of the refill behavior of patients compared to the MPR.

### 4.1. Estimating the Refill Behavior with Mean ΔT

The mean ΔT per patient showed a positive relationship with the medication possession ratio. However, a high MPR did not necessarily coincide with good refill behavior. As an example, up to 14 different ΔT mean values could be assigned to a “perfect” MPR of 1 (range 0.995–1.005, see Figure 2D). Therefore, ΔT can add valuable information to estimate medication adherence or can even be used as a more precise alternative to the MPR. Further, mean ΔT has the unit “days” and represents a more comprehensible value for researchers, health professionals, or policymakers for deciding what is an appropriate level of medication adherence compared to common CMAs. The 80% threshold to distinguish adherent from non-adherent patients [24,25,26] is mostly without clinical rationale [5,24,26] and with no precise picture of the exact patient refill behavior. The illustrative patient with an MPR of 95% had a median ΔT of 7 days and is considered as adherent according to the 80% threshold. However, patients also had gaps of −62 days during the 19 refills. In the case of non-forgiving medicines such as the DOAC where non-adherence can have fatal consequences [27], a potential 62-day gap of medication without supply can be risky at any time during the refill interval. Therefore, we question the 80% medication possession as a universally accepted threshold for good adherence [11] and suggest defining the allowable gap (that is, the negative ΔT) for a specific medicine according to a clinical rationale. 

### 4.2. Documenting the Changing Refill Behavior with ΔT

Our DOAC population had a high MPR and a positive mean ΔT, but patients were still on average 5 out of 19 times too late for their refill in the pharmacy. This provided the first indication that refill behavior was not steady over time. The trend pattern of our DOAC population was an increased oversupply of about 20% per refill (calculated with ΔT)**,** but a constant percentage of patients obtained their medication “on time” (calculated with the dichotomized form dΔT). This suggests that some DOAC patients have steadily accumulated oversupply and boosted the absolute ΔT at every refill. Even if no risky trend was observed in our population, “oversupplying” can be a critical refill behavior and has been associated with higher hospitalization rates [28,29] and increased health care costs [30]. The cluster analysis confirmed that the dominant patient group in our population were patients who refilled consistently on time or too early indicating sufficient possession of medicines. However, the cluster analysis showed that 16 patients nevertheless had gaps in their therapy. These patients present a different refill behavior that is best represented by longer breaks with no refills inserted between stable phases of sufficient supply. Different interventions are needed for these patients compared to the majority of balanced “over-suppliers”. In addition, they will be potentially missed when only applying the MPR to selected non-adherers. We chose a hierarchical cluster analysis that allowed us to show the differentiated refill behavior. Among the potentially suitable methods for forming groups, we decided to apply the cluster analysis on dΔT. By choosing the dichotomized variable, the constant oversupplies were erased, and the focus was set on patients “at clinical risk”, that is, with undersupply. Our method was able to detect 16 patients whose deviant refill behavior was dramatic and who required to be actively approached in the pharmacy. For predicting the refill behavior, group-based trajectory models could be applied to ΔT to differentiate patients into different trajectory groups. This method has been used for simulated medication adherence data [18] and real-life data [31,32]. Nevertheless, and independently of the clustering methods used, all these methods are equally useful to decide on the behavioral support the patient would need [18].

### 4.3. Potential Applications for ΔT in Research

With dispensing data, different values can be calculated to map the refill behavior of a population as a proxy for medication adherence. They are needed either for the description of population data or intervention studies. In the latter, the medication adherence estimate is usually compared before and after the intervention [33]. Hence, defining the beginning and the end of these observation periods is of paramount importance because the inclusion of medication events at the edge crucially influences the calculated CMA values (such as MPR and PDC) [8,9,10]. This source of variability is not evident with ΔT. In population studies, ΔT delivers the average refill behavior, the refill trend, or the classification into different refill groups. These estimates enable a deeper insight into the refill behavior of a patient compared to MPR. Finally, for medication synchronization and reminder programs, ΔT could be considered rather than CMAs [34] because the effect of the intervention is measured more directly by answering the question: Do patients with reminders come earlier to the pharmacy? In contrast, CMAs answer indirectly the question: Does a reminder influence the medication availability? Therefore, ΔT can depict in-depth the refill behavior of a population and can evaluate directly whether an intervention can influence refill behavior.

### 4.4. Potential Applications for ΔT in Practice

Automated real-time measurements of medication adherence exist already in pharmacy software, such as the Australian Med Screen Compliance program. These programs help community pharmacies to improve and sustain medication adherence. The system alerts the pharmacist when the MPR is below 70% and suggests an educational-based intervention [35]. Our ΔT represents a suitable calculation base for automatized medication adherence calculation systems. Because the nature of medication adherence is dynamic [36], automated calculations with ΔT are potentially more suitable to screen for non-adherent patients than the MPR, as ΔT can preset the current refill behavior of the patient. In addition, the number of theoretically remaining tablets at the refill event can be easily calculated from ΔT. This corresponds to the principle of the “pill count” adherence measurement method, which is a frequently used measurement in clinical trials [37] but not in practice [38]. Therefore, the “pill count” based on ΔT could be used to adjust the days’ supply instead of asking for feedback such as: “How many tablets have you still at home?”. Depending on the answer, a medication adherence consultation could be offered. In addition, the presented visualization of ΔT as heat map, with the intuitive traffic light scheme color code using green for refills “on time” and red for refills “too late”, could enable rapid detection of patients in need of a targeted intervention. In focus groups conducted by Fénélon-Dimanche et al. asking for the expectations of an electronic medication adherence tool based on prescription refills, the pharmacists wanted a table displaying medication adherence with a color code representing adherence level [39]. Compared to their proposed annual MPR and quarterly MPR, ΔT provides a more sophisticated mapping of patient behavior. Additionally, ΔT can be used as a quality or performance measure between pharmacies or health care plans by identifying “best practice” at a health care system level. Thus, ΔT could represent a pharmacy adherence measure that is comparable to the Pharmacy Quality Alliance measures that use the PDC [3,4].

### 4.5. Strength and Limitations

Our study has several strengths. First, ΔT is an absolute value with the unit “days” and is solely defined by the days’ supply and the refill interval. This represents a major advantage compared to established possession estimates such as the MPR or PDC that are usually reported without units as rate or percentage [8,9,10]. As an example, different values are obtained for a patient with the same alleged CMA because of different calculation methods [8,9,10], which makes comparability of studies almost impossible [19]. Second, we defined and derived ΔT from the nomenclature for CSA and CMA proposed by Steiner et al. [5], the ABC-Taxonomy [15], and the TEOS Framework [19]. With this transparent definition, we hope that researchers will not create variations of ΔT that will hinder standardization and high-quality systematic reviews in medication adherence research [40,41]. Third, we used real life data for the development of ΔT. Thus, we demonstrated that our calculations not only work in theory, but also in practice. A trend could be detected, as well as four different refill behavior groups. Fourth, the cluster analysis were applied for patients with 19 refills. This corresponds to a mean observation period of approximately three years, which supports the strength of our calculation. This study also has several limitations. First, our calculation with ΔT was focused on a single medication. Thus, the suitability of ΔT in polypharmacy remains to be shown. Variations of the MPR or PDC are already developed and used for polypharmacy [22,42,43]. Second, the mean supply duration varied between patients. This is linked to the commercially available package sizes and is an unmodifiable factor. Nevertheless, trends can be estimated. Third, the patient sample was highly selective. As expected, patients with 20 dispenses showed persistence in their therapy resulting in high medication possession rates. The largest cluster group included 71 patients who refilled their DOAC on time or too early and stockpiled with increasing refill numbers. Nevertheless, the advantage of ΔT is that even in this highly selective patient sample, patients with suboptimal refill behavior could be detected. To determine the true potential of ΔT as a useful indicator for pharmacy dispensing data, the next step should be to apply ΔT in a less selective population with a lower medication possession rate and fewer refills. Fourth, due to the selected population, the sample was small with only 116 patients. In general, CMAs are used in large EHD, which have 10 to 1000 times more patients [21,36,44]. However, it is common to use either simulation samples or small data sets when developing medication adherence measures [18,43]. Fifth, clinical data for the involved patients were sparse so that any association between the index and a clinical outcome was impossible to draw.

## 5. Conclusions

We introduced Delta T, a new absolute medication adherence measure that can characterize every refill event of a patient. Its potential applications are manifold. In practice, Delta T may support targeted, evidence-based interventions for medication adherence. In research, Delta T represents a more specific estimate compared to MPR. In our population of DOAC patients, the percentage of patients who refilled “on time” remained steady while the absolute ΔT increased, indicating few constant over-suppliers inflating ΔT. Still, it was possible with ΔT to show the dynamism in refill behavior and filter out the few patients in need of targeted medication adherence interventions. Further studies need to demonstrate the association of the new index Delta T with a clinical outcome that would require a follow-up after a targeted intervention.

## Figures and Tables

**Figure 1 pharmaceutics-14-00103-f001:**
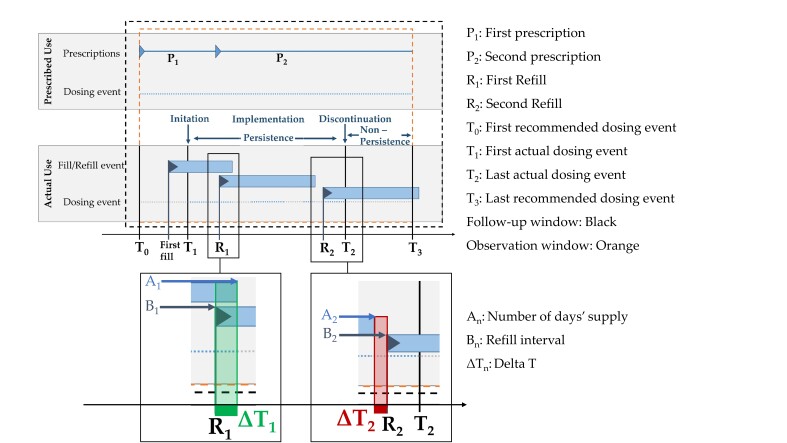
Visualization of a refill history of a fictitious patient with the phases of medication adherence according to the ABC taxonomy [11] (Adapted from Frontiers, 2018), the defined timelines and events according to the TEOS framework adapted from [15], and the characterization of the refill events with ΔT. Dots represent the dosing history with dosing event (blue) and without dosing event (grey). The backline of the triangles indicates the refill events and must not correspond to the dosing events (T0-T3). Blue bars represent the duration of the supply.

**Figure 2 pharmaceutics-14-00103-f002:**
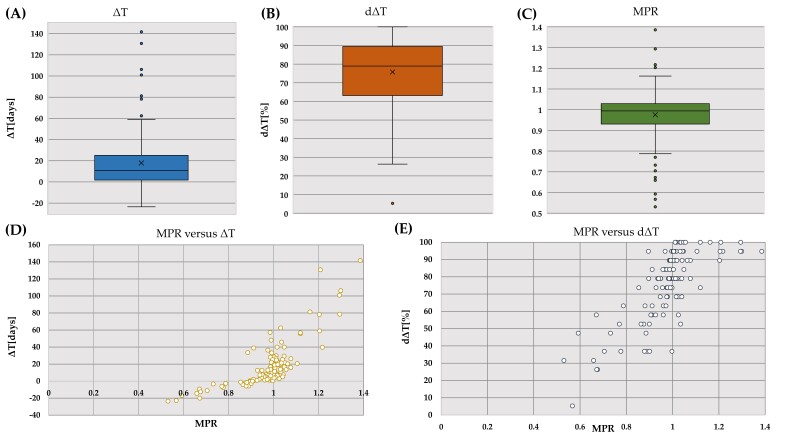
Upper panel: (**A**) distribution of ΔT around the mean of 17.8 ± 27.9; (**B**), distribution of dΔT around the mean of 75.8 ± 20.2%; (**C**) distribution of MPR around the mean of 0.975 ± 0.129 lower panel: (**D**) ΔT graphed against the corresponding MPR; (**E**) mean dΔT graphed against the corresponding MPR. MPR, medication possession ratio; ΔT, mean ΔT; dΔT, mean dichotomized ΔT. Refer to text for details.

**Figure 3 pharmaceutics-14-00103-f003:**
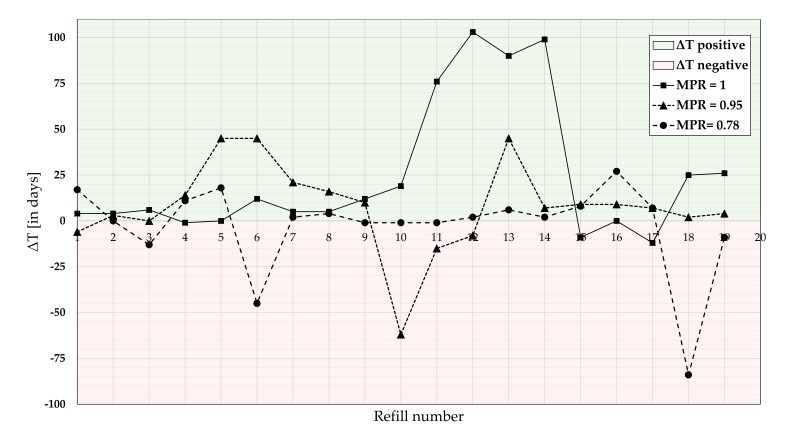
Unique refill patterns with ΔT from three illustrative patients with MPR of 1 (square), of 0.95 (triangle), and of 0.78 (dots) over the period of 19 refills. The green area above the x-axis indicates positive ΔT that is, refilling too early and oversupply; the red area below the x-axis indicates negative ΔT that is, refilling too late and gaps in DOAC supply.

**Figure 4 pharmaceutics-14-00103-f004:**
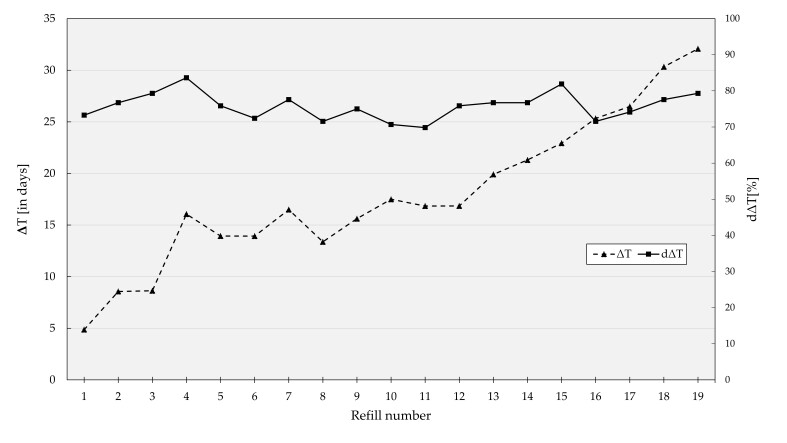
Scatter plots of ΔT (triangle) and dΔT (square) against the refill number over the period of 19 refills; *y*-axis in days (**left**) and percent (**right**).

**Figure 5 pharmaceutics-14-00103-f005:**
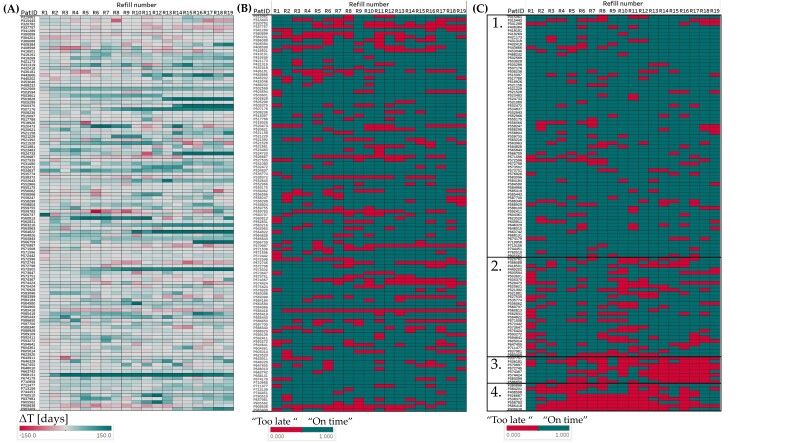
Heat maps replicating all 2204 ΔT (19 refills × 116 patients) per refill number with green color indicating refill events “on time” (ΔT ≥ 0) and red color indicating events “too late” (ΔT < 0); (**A**) with a color gradient from +150 days (green) to −150 days (red); (**B**) after dichotomization into green (“on time”) and red (“too late”); and (**C**) after clustering into the 4 refill groups: (1) refills are mostly “on time”, (2) erratic refills, (3) gaps at the end of the refill period, and (4) gaps in the middle of the refill period.

**Table 1 pharmaceutics-14-00103-t001:** Values of ΔT for the three illustrative patients.

Patient	Median ΔT (IQR) [In Days]	Mean ΔT ± SD [In Days]	Max ΔT [In Days]	Min ΔT [In Days]	Range (=Max ΔT–Min ΔT) [In Days]	dΔT [%]	Sum of Days without Supply (=Sum of Negative ΔT) [In Days]
MPR = 1	6 (26)	24.4 ± 36.5.3	103	−12	115	84.2	22
MPR = 0.95	7 (16)	7.7 ± 23.4	45	−62	107	78.9	91
MPR = 0.78	2 (9)	−2.6 ± 23.9	27	−84	111	63.2	154

**Table 2 pharmaceutics-14-00103-t002:** Characteristics of the four refill groups after clustering of the 2204 ΔT values obtained from 19 refills from 116 patients.

Cluster Number	Characterization of the Clusters	Number of Patients (%)	Mean Age ± SD [In Years]	Percentage of Women [%]	ΔT ± SD [In Days]	dΔT ± SD [%]	MPR ± SD
1	Refills “on time”	71 (61.2)	70.8 ± 10.3	54.9	25.4 ± 27.7	87.3 ± 10.2	1.01 ± 0.09
2	Erratic refills	29 (25.0)	75.1 ± 12.3	54.7	19.0 ± 29.7	72.1 ± 13.3	0.99 ± 0.10
3	Gaps at the end of refill period	8 (6.9)	70.1 ± 13.7	37.5%	−6.6 ± 7.8	39.5 ± 8.9	0.79 ± 0.13
4	Gaps in the middle of refill period	8 (6.9)	71.1 ± 12.3	62.5%	−12.0 ± 8.9	34.9 ± 15.1	0.75 ± 0.13

## Data Availability

The data presented in this study are available upon request from the corresponding author.

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
