# Peer review of "Delta T, a Useful Indicator for Pharmacy Dispensing Data to Monitor Medication Adherence"

_pharmaceutics, 2022, doi:10.3390/pharmaceutics14010103_

Round 1
Reviewer 1 Report
General comments:
- sometimes, the space before the "[" is missing (e.g., Arnet[18], calculated[7], DOACs)[23])
- do not use abbreviation in the titles and subtitels.
Abstract:
- define MP abbreviation.
- Methods: when? where? how the cases were selected? whcih were the inclusion and exclusion criteria? etc.
- 2’204? is this 2204?
Keywords: "cluster analysis," is the comma necessar?
Introduction:
- How the CMA and PDC are interpreted?
- "different medication adher-ence rates can be calculated with the same patient data" such as? which are their usefulness?
- Present the suefulness of the new proposed approaches.
- "We applied the novel adherence estimate in a real-life dispensing data set to visualize it, to compare it to the traditional possession measure MPR, and to draw recommendations for future work." this belongs to the Methods section.
Methods:
- "Positive values (ΔT ≥0) indicate the number of days the patient has an “oversupply”" ... it is not correct to say oversupply when ΔT = 0
- why this gap between ethics committee approval and disemination of results?
- It could be expected to have different behavior now? (discuss in the limitation of the study if apply)
- "we selected a sample of patients of persistent DOAC users with at least 20 consecutive" how?
- The subsection that describe statistical analysis is missing.
Results:
- "2’919"?
- The characteristics of the evaluated patients is missing. A different behavior is expected according to gender and age.
- define range ( min to max or Q1 to Q3). Put the ifnormation in Methods section.
- since "dΔT" is a dicotomial variable does not accomplish the criteria for linear regression analysis.
- As the scatter show in Figure 2 the better relationship is not linear. Furthermore, if ΔT does not follow the normal distribution the assumption of normality of dependent variable is broken so the linear regression is not appropriate.
- It is expected to compare the performances of the proposed approach with an existing approach.
Discussion:
- "We introduced Delta T (ΔT) as a proxy to medication adherence with information on its dynamic. ΔT is an absolute value calculated from patients’ dispensing data and has the unit “days”. It allows to define every refill event as being “on time” or “too late”. In addi-tion, ΔT takes oversupply from one interval to the next into account and mimicks a prob-able real-life situation that is, the fact that patients start a new packaging when the older one is finished. " this is duplicated information.
- "It is useful to begin the discussion by briefly summarizing the main findings, and explore possible mechanisms or explanations for these findings. Emphasize the new and important aspects of your study and put your findings in the context of the totality of the relevant evidence. State the limitations of your study, and explore the implications of your findings for future research and for clinical practice or policy. Discuss the influence or association of variables, such as sex and/or gender, on your findings, where appropriate, and the limitations of the data. Do not repeat in detail data or other information given in other parts of the manuscript, such as in the Introduction or the Results section. " (http://www.icmje.org/recommendations/browse/manuscript-preparation/preparing-for-submission.html#f)
Conclusions:
- "Link the conclusions with the goals of the study but avoid unqualified statements and conclusions not adequately supported by the data. In particular, distinguish between clinical and statistical significance, and avoid making statements on economic benefits and costs unless the manuscript includes the appropriate economic data and analyses. Avoid claiming priority or alluding to work that has not been completed. State new hypotheses when warranted, but label them clearly. " http://www.icmje.org/recommendations/browse/manuscript-preparation/preparing-for-submission.html#f
Reviewer 2 Report
I’ve read with attention the paper of Baumgartner et al. that is potentially of interest. The background and aim of the study have been clearly defined. The authors tested a new adherence estimate ΔT that characterizes each refill event and depicts the dynamics of adherence. The methodology applied is overall correct, the results are reliable and adequately discussed. The only limitation of this study is the poor clinical characterization of the involved patients and an association of the index with a clinical outcome (that would of course require a very longer follow-up). This could be shortly commented
Round 2
Reviewer 1 Report
Your manuscript looks better now.